# Crack Detection and Crack Length Measurement with the DC Potential Drop Method–Possibilities, Challenges and New Developments

**Jürgen Bär**

Institute of Materials Science, Universität der Bundeswehr München, D-85577 Neubiberg, Germany; juergen.baer@unibw.de; Tel.: +49-89-6004-2561

**Abstract:** The direct current potential drop method (DCPDM) is widely used to determine the crack length in fatigue experiments. In practice, some special features of this method must be considered. Aside from general information on the experimental setup and calibration, some special features of the method, such as the influence of the ambient atmosphere and the application of the method to ferromagnetic materials, are presented and discussed. In addition, with the multiple potential drop measurement, a method is presented which improves the resolution of the DCPDM for detection of cracks and allows to determine crack initiation sites. The capabilities provided by this method are demonstrated on the basis of measurements undertaken on notched round bars and single edged notched specimens.

**Keywords:** potential drop measurement; fatigue; crack; crack initiation

## 1. Introduction

The failure of metallic materials under fatigue loading conditions is governed by the initiation and propagation of cracks. To investigate the fatigue behavior, different methods are used to measure crack lengths. The optical measurement with a travelling microscope [1,2] gives only the crack length on one surface and can only be automated with great effort. Aside from the light optical measurement of crack length, infrared thermography [3,4] and digital image correlation [5] are also used. However, these methods require special and expensive equipment and afford a complicated evaluation. An automated integral measurement of the crack length, especially for compact tension and single edge notched bending-specimen, can be undertaken by measuring the crack opening displacement [6,7]. Since the 1950s the direct current potential drop method (DCPDM) has been widely used to determine the crack length in fatigue experiments on specimens [8–14] and components [15]. The DCPDM can easily be automated and thus enables crack propagation experiments under stress intensity controlled conditions to be carried out automatically [16,17]. The alternating current potential drop method (ACPDM) uses the skin effect which is why this method is very sensitive to changes at the specimen surface and is suitable for detection of small surface cracks only. Accordingly, the ACPDM is more complicated to use and requires expensive equipment [18,19]. Although the DCPDM is less sensitive to short crack propagation, it was successfully used to detect crack initiation in fatigue experiments [20,21].

In this paper, some practical aspects of crack length measurement in fatigue tests on metallic materials will be presented and discussed. In the second part of the article, the benefits of a multiple potential drop measurement for the detection of cracks will be presented. This method offers an improved resolution compared to a single measurement and also enables the localization of the crack initiation site.

## 2. Crack Length Measurement with the DC-Potential Drop Method

*2.1. Experimental Setup*

The crack length measurement with the DC potential drop method requires only a very simple experimental setup. As shown in Figure 1a, a constant, time-stable direct current is passed through the sample. The potential is measured between two grips arranged at the specimen in a distance $y_0$ to the notch root. A good connection between the grips and the specimen is essential for a stable measurement. When screws or pressed-in pins are used, the resistivity of the connection may vary during cyclic loading leading to fluctuations in the measured potential values. This can be avoided by directly spot-welding the grips to the specimen surface [13,22]. Due to the good conductivity of metallic materials, high currents as well as sensitive voltmeters are needed to measure a reasonable potential drop with adequate accuracy. The use of highly sensitive amplifiers in the control electronics of the testing machine allows a simple and simultaneous measurement of all data like force, stroke, and potential [23,24].

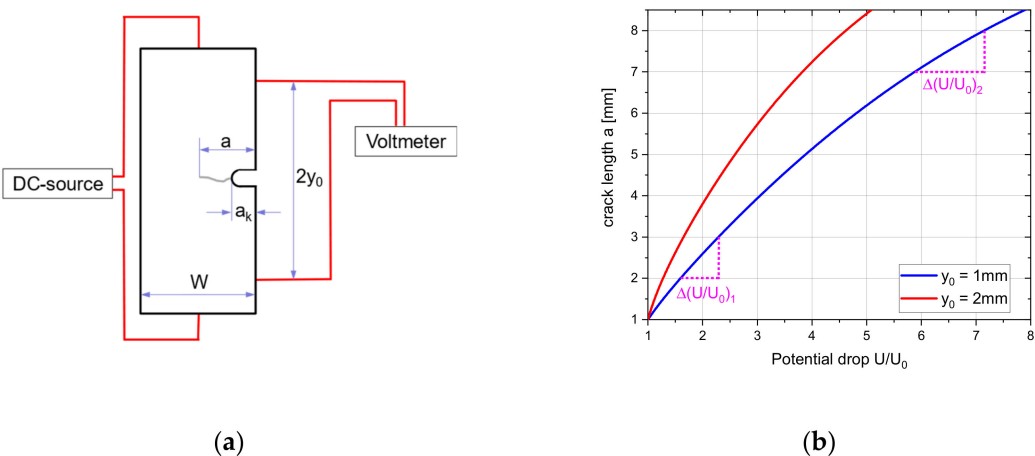

(**a**)　　　　　　　　　　　　　　　　　　　　　　　　　(**b**)

**Figure 1.** (**a**) Schematic drawing of the setup of the direct current potential drop (DCPD) measurement; (**b**) the resolution of the direct current potential drop method (DCPDM) depends on the distance of the potential grips $2y_0$ and the crack length a.

Johnson [8] developed a calibration function for center crack tension (CCT) and single edge notched (SEN) specimen to calculate the crack length from the measured potential drop. The same result was achieved by Gilbey and Pearson [25] using a different mathematical approach. For SEN specimen, the crack length can be calculated by the so-called Johnson equation (Equation (1)):

$$a = \frac{2 \cdot W}{\pi} \cdot arccos\left[ \frac{cosh \frac{\pi \cdot y_0}{2 \cdot W}}{cosh\left( \frac{U}{U_0} \cdot arcosh \frac{cosh \frac{\pi \cdot y_0}{2 \cdot W}}{cos \frac{\pi \cdot a_K}{2 \cdot W}} \right)} \right], \tag{1}$$

The crack length calculated from the potential drop $U/U_0$ for a SEN-specimen with a width of 12 mm is shown in Figure 1b. The curves show that the resolution of the DCPDM is rising with crack length. Additionally, the sensitivity is influenced by the distance of the potential grips $y_0$. To achieve a high resolution, the distance of the grips should be minimized. Unfortunately, this leads to a reduction of the measured potential, while the noise of the voltmeter or amplifier used for the potential measurement becomes more and more dominant and limits the accuracy of the measurement. Therefore, the optimal distance of the potential grips is always a compromise between these two opposing effects.

## 2.2. Calibration of the Crack Length Measurement

The Johnson formula (Equation (1)) as an analytical solution of Laplace's differential equation contains, apart from the potential ratio $U/U_0$, only geometric quantities and should provide exact crack lengths when using the exact specimen dimensions. In crack propagation experiments, however, there are clear discrepancies between the crack lengths determined on the fracture surface and the potential drop measurement. For this reason, a calibration of the DCPDM, i.e., an adaptation of the Johnson equation, is necessary. Aside from a calibration using finite element method (FEM) calculations which is suitable for special crack geometries [12,26], the calibration can be undertaken by simulating the crack propagation via defined saw cuts and correlating the measured potential probe data with the "artificial" crack lengths generated this way [27,28]. Another possibility is to mark the crack extension on the fracture surface by overloads or by load cycle blocks with a different stress ratio, so-called marker loads [22,27–29]. Using saw cuts has the advantage that the "crack front" runs almost perpendicular to the direction of crack propagation. In case of saw cuts, the measurement of the "crack length" on the surface of the specimen is sufficient, while for fatigue cracks, the crack length is underestimated due to a curvature of the crack front.

For the calibration of the DCPDM, the marking of the crack length by means of overloads as well as marker loads has proven itself. A suitable method to equalize the curvature of the crack front is the measurement of the crack area. The mean crack length is obtained by dividing the measured crack area by the specimen width. For a SEN-specimen made of AA7475, the crack fronts marked by overloads are shown in Figure 2a as red lines and the resulting mean crack lengths perpendicular to the propagation direction are shown as yellow dashed lines. Regarding this, a linear dependency of potential and crack length is assumed, whereby the error in the case of curved crack fronts can only be minimized, but not eliminated [30,31]. The calibration is undertaken by drawing the optical crack length a divided by the specimen width (w) against the measured potential drop $U/U_0$ and adapting the Johnson equation with a least square fit by using the distance of the potential grips $y_0$ as a free parameter. As a result, a grip spacing of $y_0 = 2.53$ mm was obtained. The curve for the geometrical grip spacing of $y_0 = 2$ mm added for comparison in Figure 2b shows clearly that the actual crack length is underestimated when using the real geometric spacing of the potential grips.

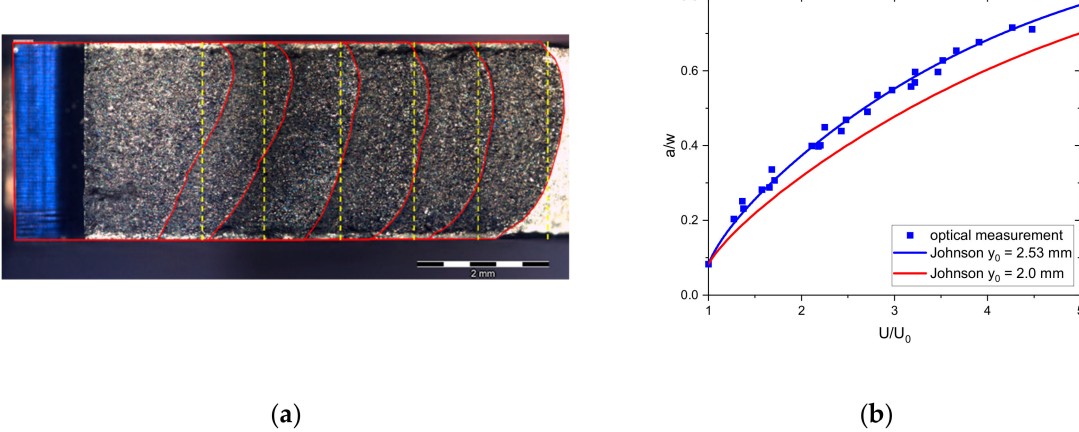

(**a**)          (**b**)

**Figure 2.** (**a**) Crack surface with overload marks (red lines). For calibration of the DCPDM, the cracked area is measured and divided by the specimen thickness, resulting in mean crack length (yellow lines); (**b**) the effective distance of the potential grips $y_0$ is achieved by a least square fit of the Johnson equation (Equation (1)) on the measured optical crack length a divided by the specimen width (w).

## 2.3. Influence of Temperature and Atmosphere

The DCPDM can be influenced by several external effects. For example, a change in temperature causes a change in electrical resistance and thus an increase in the measured potential. Because of the

low potential values, the temperature must be kept as constant as possible during the experiment. To avoid this problem, Ljustell [32,33] uses temperature compensation implemented by means of a reference potential drop measurement on an unloaded sample. However, the main problem when servohydraulic testing machines are used is not the change of the ambient temperature, but the heating of the oil of the testing machine. These heating effects can easily be avoided by a temperature stabilizing period prior to the experiment. Within this period, the temperature of the complete system is levelled out and therefore no undesirable temperature effects in the experiment must be feared.

Furthermore, other effects are known which might also influence the potential probe measurement. A touching of the fracture surfaces behind the crack tip leads to a formation of electrical contact bridges. This results in a reduction in electrical resistance and, accordingly, a decrease in the measured potential leading to a shortening of the measured crack length [27,31,34]. This problem can be avoided by only using the maximum value of the potential within a cycle to determine the crack length since, at least under mode I loading, it can be assumed that there are no contact bridges under maximum loading [22,35].

In case of aluminum alloys, this effect is of minor importance due to the rapid formation of an insulating oxide layer on the crack surface, so that the differences between maximum and minimum values within a cycle are very small. Crack propagation measurements under symmetrical tension-compression loading on a SEN-specimen of a particle-reinforced aluminum alloy in dry nitrogen atmosphere, however, show an extreme change of the potential within a cycle due to the lack of oxide layer formation on the fracture surfaces (Figure 3) [36]. The measured potential varies more than 0.3 mV within a cycle which corresponds to a change in the crack length of nearly 3 mm. This example impressively underlines the need to use the maximum value of the potential to determine the crack length.

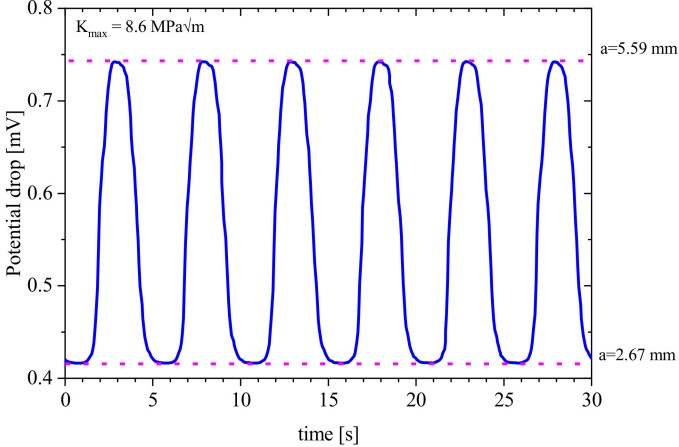

**Figure 3.** Change of the potential in a particle reinforced 6113 aluminum alloy tested under fully reversed loading conditions (R = −1) in dry nitrogen [36]. Due to the absence of oxygen, the crack flanks are not isolated and the crack length changes about 3 mm within a cycle.

### 2.4. Investigation of Ferromagnetic Alloys–The Villari Effect

Another effect influencing potential drop measurements on cyclic loaded ferromagnetic alloys is the Villari effect [37]. This effect, also known as inverse magnetostriction, is based on a change in the magnetic properties when a mechanical load is applied. The mechanical loading induces a current and thus a voltage in the sample [38]. This voltage depends on the rate of stress change $d\sigma/dt$ and consequently, on the loading frequency and level.

The potential changes of a crack free SEN specimen, made of an unalloyed steel (C45E) and loaded mechanically at a frequency of f = 20 Hz with an amplitude of 2 kN and 5 kN, are shown in Figure 4a,b, respectively. For better comparability, the same scaling has been selected in both diagrams. While only slight fluctuations in the potential can be seen at an amplitude of 2 kN, a load with an amplitude of 5 kN results in sinusoidal potential fluctuations with an oscillation range of approximately 4 μV.

The force and potential curves are out of phase. The maximum and the minimum of the potential curve are located in the region of the greatest steepness of the loading curve, i.e., at the highest stress change rate $d\sigma/dt$ at the zero crossing of the stress curve.

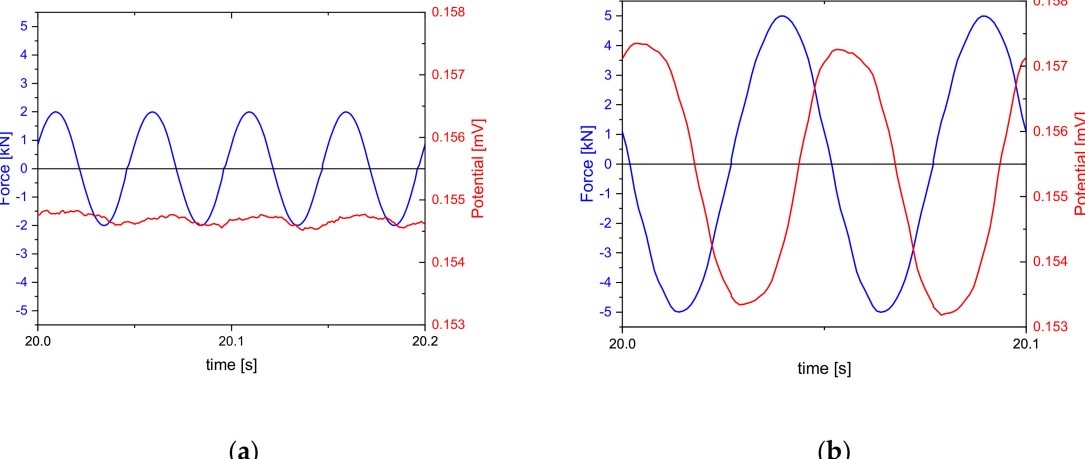

(**a**)                                      (**b**)

**Figure 4.** Change of the potential in an unalloyed ferritic steel (C45E) at loading frequencies of 20 Hz: (**a**) at low loading the potential remains nearly constant; (**b**) at higher loadings, a distinct change of the potential due to the Villari effect is visible. The maximum and minimum of the potential appear at the zero-crossing of the force where the change of the force per time is highest.

In the case of samples with cracks, the superposition of crack opening effects and the Villari effect results in more complex potential hysteresis loops. A direct comparison of the potential curve of a sample made of ferritic steel C45E and one made of austenitic steel X5CrNi18-10 with a crack length of about 6 mm each is shown in the Figure 5a. The ferritic steel shows a complex potential curve that is caused by the superposition of the crack opening that occurs synchronously with the load and the phase shifted Villari effect. The potential changes caused by the Villari effect are significantly higher than the effects caused by crack opening and closing. In Figure 5b, the corresponding hysteresis loops are shown. The austenitic steel has the maximum and minimum potential values at the force maximum and minimum, whereas the ferritic steel exhibits a completely deformed hysteresis loop with maximum and minimum values for the potential far away from the maximum and minimum loading.

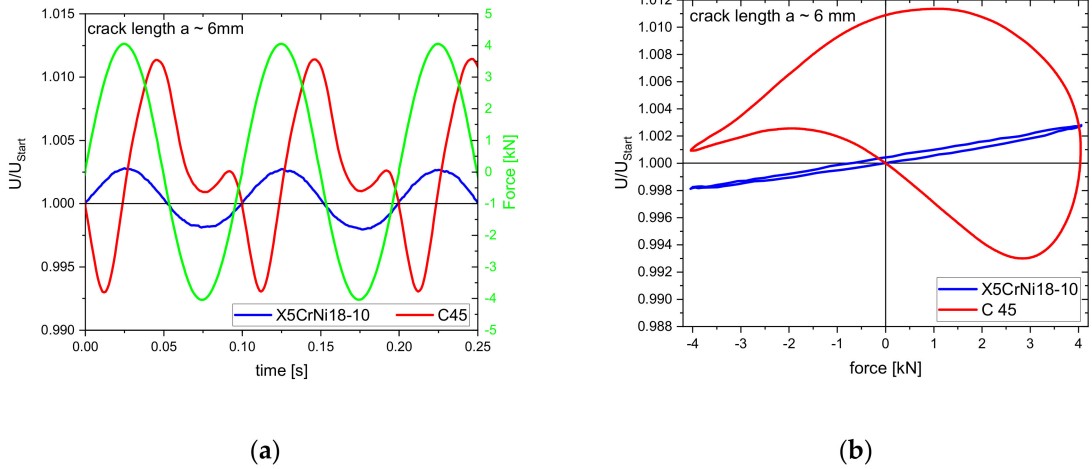

(**a**)                                      (**b**)

**Figure 5.** Changes of the potential in a specimen of an austenitic (X5CrNi18-10) and a ferritic (C45E) steel with cracks of about 6 mm length: (**a**) the austenic steel shows only a change due to the crack opening and closing, the ferritic steel exhibits two maxima, the higher one is due to the Villari effect; (**b**) corresponding hysteresis loops showing the big potential change due to the Villari effect.

In addition to the frequency and the load level, the crack length also influences the magnitude of the potential changes caused by the Villari effect. Figure 6 shows the potential change within a cycle measured on SEN specimens made of C45E and X5CrNi18-10 tested at a test frequency of 10 Hz under fully reversed cyclic loading with a load amplitude of 4 kN as a function of the crack length. The austenitic steel shows a potential difference that increases slightly with increasing crack length. This is due to the contact of the crack flanks under compression loading, where the contact area increases with the crack length. In the case of the ferromagnetic sample made from C45E, the increase in the potential difference is much steeper. This increase cannot be explained by the contact of the crack flanks alone, it is rather an increasing potential difference due to the Villari effect. Davis and Plumbridge [37] observed the same effect in their investigations. This increase in the potential difference can be attributed to the stresses at the crack tip increasing with the length of the crack, and the magnetic properties that depend on the crack length [39].

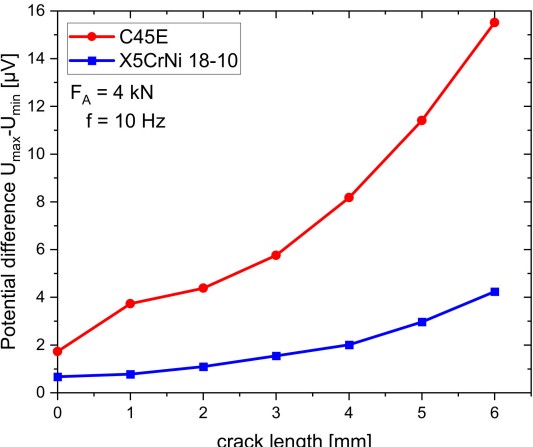

**Figure 6.** Potential differences for a high alloyed steel (X5CrNi 18-10) and a ferritic steel (C45E). With increasing crack length, the measured potential difference rises. The ferritic steel shows a steeper increase due to the Villari effect.

The Villari effect can cause problems in testing ferromagnetic materials when the maximum potential within a cycle is used to calculate the crack length. If, for example, a test according to ASTM E647 [6] is carried out, the load level is reduced to reach the threshold value. Since the potential change induced by the Villary effect increases with the crack length and the loading level, the influence of this effect on the measured potential changes during the experiment. This error can be avoided if the potential at the maximum force is used to calculate the crack length. Since this value is not available by the control electronics and can only be extracted from the hysteresis loops, it is advisable to use the mean value of the potential. However, it must be noted that, as mentioned in Section 2.3, measurement errors may occur due to contacts of electrically conductive fracture surfaces.

Despite the problems presented here, which must always be taken into account when evaluating the results, the DCPDM is the most suitable and preferred method for crack length measurement in electrically conductive materials due to its high accuracy and ease of automation. In order to reduce or even avoid measurement errors, the special features of the examined material must be considered. The potential probe hysteresis should be used for this purpose, as these are the best way to identify the peculiarities that occur during the measurement.

## 3. Crack Detection and Crack Localization with Multiple PD Measurements

As a first approximation, the PD measurement is based on a reduction in area caused by the propagating crack and an associated increase of the electrical resistance. In fact, the potential drop measurement bases on a densification of the potential field lines in front of the crack tip [40]. For this

reason, the position of the potential probe is of crucial importance [9,22]. Bär and Tiedemann [12] showed that, especially in the case of short cracks at notches, the shape and the location of the initiated crack has an essential influence on the measured potential values. The investigations by Campagnolo et al. [24] on notched round bars also showed a distinct influence of the position of the crack and the crack geometry on the level of the measured potential. Hartweg and Bär [23] showed that the position of the crack initiation site on the circumference of a notched round bar can be reliably determined with a multiple potential drop measurement. In the following, the possibilities offered by multiple potential drop measurement for determining the crack position and the detection of short cracks in notched round bars and single edge notched specimens will be presented.

### 3.1. Crack Detection in Round Bars

The investigations for crack detection on notched round bars were carried out on specimens of a high-alloy steel (X8CrNi18-9) with a diameter of 19.8 mm and a circumferential notch with a radius of 5 mm and a depth of 3.9 mm as shown in Figure 7a. To measure the potential, wires with a diameter of 0.2 mm were spot-welded to the specimen surface at a distance of 0.5 mm from the upper or lower edge of the notch (Figure 7b). The three measuring points were equidistantly arranged along the circumference of the sample with an angle of 120° in between two points, as shown in Figure 7c.

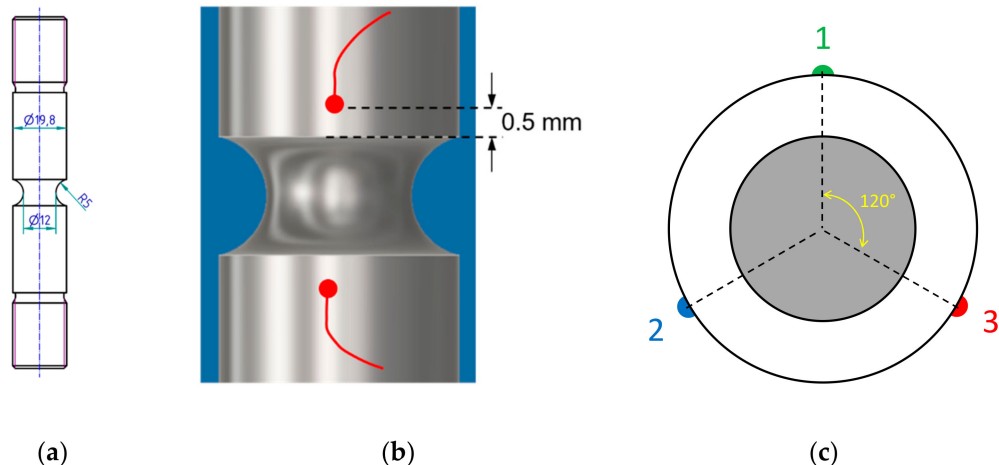

(a) (b) (c)

**Figure 7.** Setup of notched bar for multiple potential drop (PD) measurement: (**a**) drawing of the specimen; (**b**) position of the potential grips beside the notch; (**c**) position of the potential grips on the circumference.

The tests were carried out under fully reversed loading conditions with an amplitude of 35 kN and a frequency of 20 Hz. To mark the crack front on the fracture surface, overloads with a maximum load of 70 kN were applied at intervals of 5000 cycles, beginning after 10,000 cycles. In order to initiate cracks at defined positions, secondary notches in form of small laser-cuts were introduced into the notch root.

A constant current was conducted via the clamps through the specimen. The three potentials were measured using the amplifiers of the control electronics (Doli EDC 580V). During the fatigue tests, the maximum and minimum value of the three potentials were recorded for each cycle. In order to compensate differences in the potentials due to small differences in the tap spacing and contact resistance, the relative potentials $P_i$ were calculated as the quotient of the measured potential $U_i$ and the mean value of the potential of the first 10 cycles $U_{i,0}$ according to Equation (2):

$$P_i = \frac{U_i}{U_{i,0}}, \qquad (2)$$

Figure 8a shows the fracture surface of a specimen with an additional laser cut at the position of potential 1. The overload lines are retraced in color on the image of the fracture surface. The run of the three relative potentials $P_1$, $P_2$, and $P_3$ as a function of the crack area related to the sample cross-section is also shown in Figure 8a. The relative potential $P_1$ shows a steeper increase compared to the values of $P_2$ and $P_3$, which show an almost identical increase. This clearly shows that the measured potential depends on the position of the potential probe and thus multiple potential measurements can be used to determine the location of crack initiation.

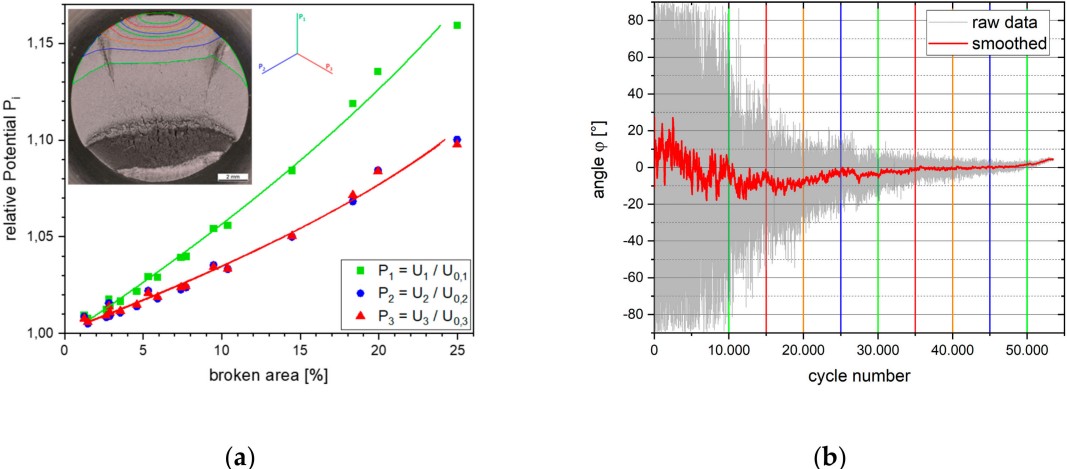

(**a**)                                                                   (**b**)

**Figure 8.** Results of an experiment with a crack initiation at the position of potential probe 1. The crack front was marked by introducing overloads (colored lines); (**a**) the relative potential $P_1$ shows a steeper increase compared to the values of $P_2$ and $P_3$; (**b**) the angle calculated with the geometrical model shows a high scatter in the beginning of the experiment. After crack initiation the scatter decreases, and the position of the initiated crack can be determined. The colored lines correspond to the crack extensions shown in the crack surface in Figure 8a.

Hartweg and Bär [23] developed a simple geometric model to locate the crack initiation site based on the three measured potentials. In this model, the relative potentials are treated as vectors that span a plane. In the crack-free sample, all relative potentials are equal and show the value $P_i = 1$. This results in a horizontal plane and the normal vector of this plane is parallel to the loading axis. When a single crack is initiated on the specimen circumference, the relative potentials $P_i$ begin to differ and therefore the plane and thus the corresponding normal vector tilts in the direction away from the crack initiation site. The position of the crack in relation to potential probe $P_1$ can be described by the angle $\phi$ which is the opposite direction of the deflection of the normal vector.

The angle $\phi$ calculated from the potential data for the specimen from Figure 8a is shown in Figure 8b as a function of the cycle number. The scatter of the unfiltered raw data decreases with increasing cycle number and thus crack size. The scatter has already decreased significantly after 10,000 cycles. At this point the crack has propagated up to 1.48% of the total cross-sectional area. After 30,000 cycles with a relative crack area of 5.92%, the standard deviation is below 5°. Both the raw data and the angle values, smoothed with a moving average over 100 points, indicate the formation of a crack at an angle of 0°, i.e., at potential probe $P_1$. The reduction in scatter of the angle $\phi$ seems to be a suitable criterion for crack initiation.

In Figure 9 the data of an experiment without a temperature stabilization prior to the experiment are shown. At the beginning of the experiment a steep increase of all three relative potentials due to a heating-up of the specimen is visible (Figure 9a). A first crack on the fracture surface near potential probe 3 occurs after 115,000 cycles. After that point, a second increase of all potentials takes place, where potential 3 shows the steepest increase. The angle $\phi$ shows a pronounced scatter around a nearly constant mean value up to about 120,000 cycles, thereafter the scatter decreases significantly,

and the mean value indicates the crack formation at an angle of about 103°. The following change of the mean value indicates that the further crack propagation is mainly in direction of higher angles and then again to lower angles which corresponds well with the observation on the fracture surface.

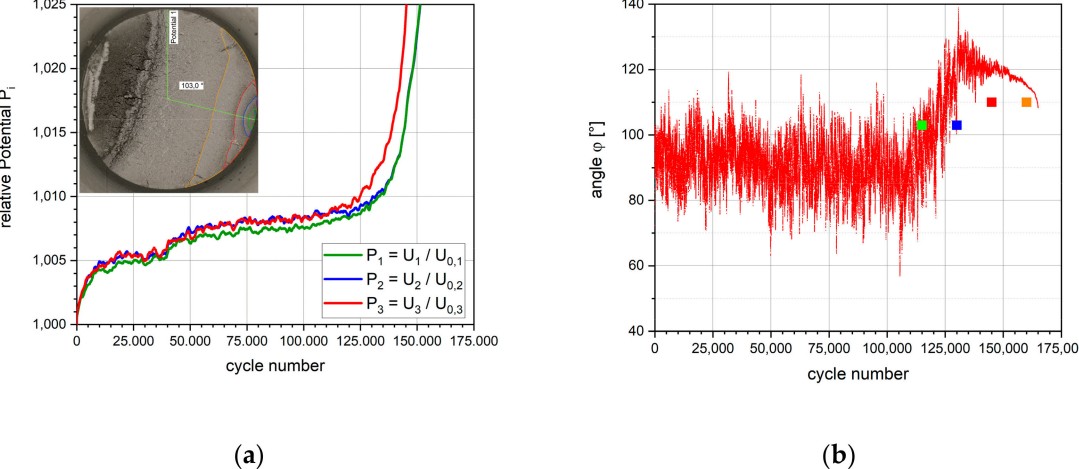

(**a**)　　　　　　　　　　　　　　　　　　　　　　　(**b**)

**Figure 9.** Data of an experiment without prior temperature stabilization: (**a**) run of the three relative potentials and the crack surface with marked crack extensions; (**b**) calculated angle φ showing a crack initiation after about 110,000 cycles. The crack was initiated at an angle φ of 103°, the green square at N = 115,000 corresponds to a cracked area of 1.27%, the blue square at N = 130,000 to 4.8%.

The measurements show that multiple potential drop measurements on notched round specimens evaluated with a simple geometric model allow the determination of the crack location even for relatively short cracks. The scatter of the calculated angles can be used as a criterion for crack detection. On the one hand, this method improves the sensitivity of the DCPDM in the case of short crack lengths and eliminates potential changes due to temperature fluctuations. On the other hand, however, the method presented here only provides reliable values for the location of single cracks. In case of multiple cracks, only the center of gravity of the cracks is displayed by the geometric model.

### 3.2. Crack Detection in Single Edge Notched Specimens

The investigations were carried out on SEN-specimens with a size of 80 × 12 × 2.8 mm made of aluminum alloy 7475 T761. The potential grips were spot-welded in a distance of 1.5 mm from the notch root. As shown in Figure 10a, potential probes were attached to the front ($U_F$), the rear ($U_B$), and the narrow ($U_N$) side of the sample. Figure 10b shows an optical micrograph of the spot-welded wires beneath the notch.

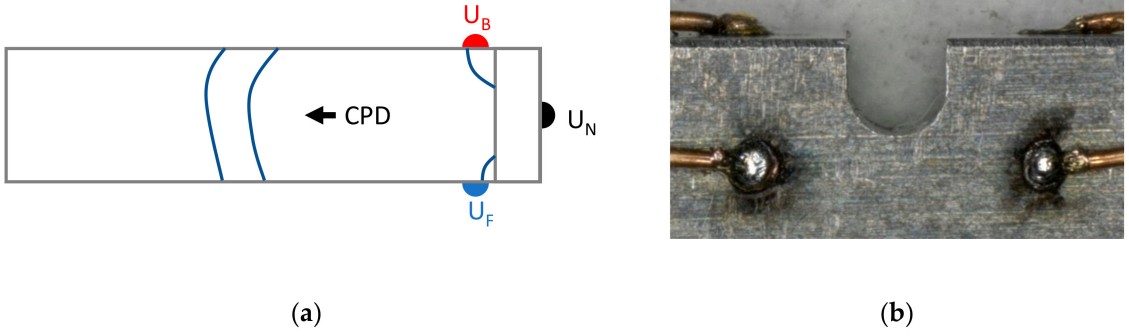

(**a**)　　　　　　　　　　　　　　　　　　　　　　　(**b**)

**Figure 10.** Single edge notched (SEN)-specimen: (**a**) position of the potential grips on the specimen. The notch is located on the right side, the arrow characterizes the crack propagation direction (CPD); (**b**) micrograph of the laser-welded wires on the specimen surface.

The specimens were cyclically loaded with a stress amplitude of 70 MPa and a frequency of 20 Hz under fully reversed loading conditions in a testing machine with parallel clamps. At intervals of 15,000 load cycles, overloads with an amplitude of 175 MPa were applied to mark the actual crack geometry on the fracture surface. The machine control and data acquisition were undertaken by a control electronics of the type Doli EDC 580 V.

The measured potentials $U_F$, $U_B$, and $U_N$ were normalized according to Equation (2) to the mean value of the first 10 cycles $U_{F,0}$, $U_{B,0}$, and $U_{N,0}$, respectively. From the relative potentials $P_F$, $P_B$, and $P_N$ determined in this way, the quotients $Q_{Front}$ and $Q_{Back}$ were calculated according to Equation (3):

$$Q_{Front} = \frac{U_F}{U_N}, \text{ and } Q_{Back} = \frac{U_B}{U_N}, \tag{3}$$

For an uncracked specimen at the beginning of the experiment, both quotients exhibit the value $Q_{Front} = Q_{Back} = 1$. When a crack is initiated, a change of the potential quotients depending on the crack initiation site takes place. In Figure 11, the possible changes of the potential quotients for three idealized crack initiation scenarios are sketched:

- Figure 11a: When a crack is initiated near the frontside (backside) $Q_{Front}$ ($Q_{Back}$) increases while $Q_{Back}$ ($Q_{Front}$) decreases or remains constant.
- Figure 11b: When a crack is initiated in the center of the notch root a decrease of both potential quotients takes place.
- Figure 11c: When cracks are initiated on the front and backside at the same time, both quotients are rising.

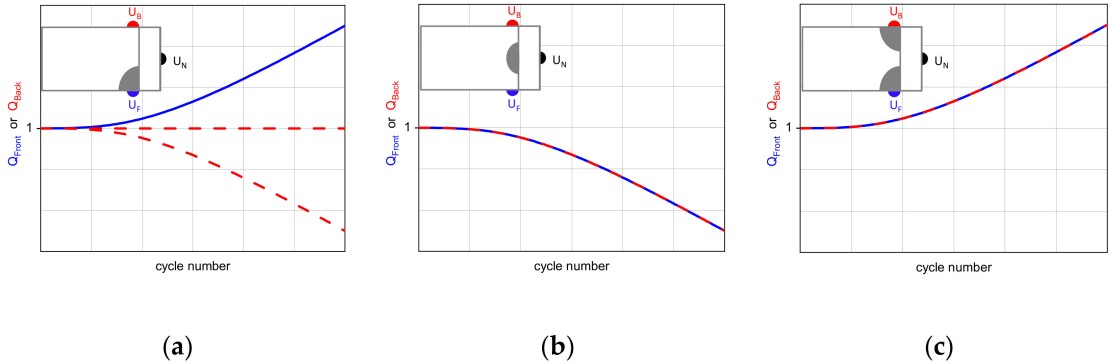

(a)　　　　　　　　　　　　　(b)　　　　　　　　　　　　　(c)

**Figure 11.** Run of the potential quotients $Q_{Front}$ and $Q_{Back}$ for different crack initiation sites: (**a**) single crack on the front side; (**b**) single crack in the middle of the notch root; (**c**) two cracks on the front and the backside.

The formation of subsequent cracks can also be described based on the change of the potential quotients described above. In the following, this described criteria shall be applied on real experimental data.

Figure 12a shows the fracture surface and the corresponding run of the potential quotients $Q_{Front}$ and $Q_{Back}$ of a sample with a crack initiated near the backside. The yellow lines show the crack fronts marked with overloads in an interval of 15,000 cycles. The marks of the first overload show a crack with a length of about 0.9 mm and a depth of 0.25 mm. The formation of the crack is shown by an increase of the potential quotient $Q_{Back}$ and a decrease of the quotient $Q_{Front}$. With further crack propagation the crack front becomes more and more straight and parallel to the notch and consequently the potential ratios are increasingly converging.

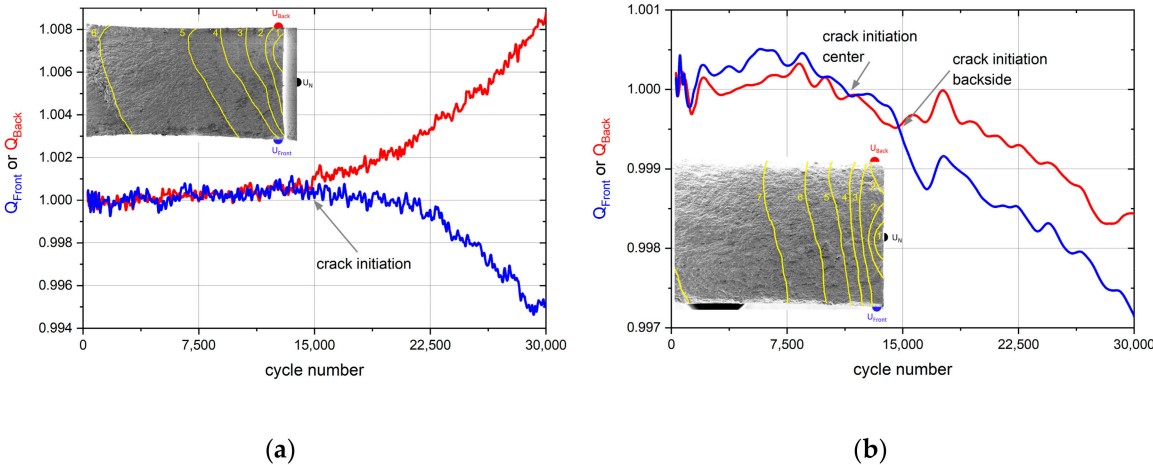

**Figure 12.** Curves of the potential quotients for different crack initiation sites in SEN-specimens. The crack front was marked in intervals of 15,000 cycles by introducing overloads (yellow lines): (**a**) crack initiation near the backside; (**b**) crack initiation in the specimen center with a secondary crack at the backside.

In a further sample shown in Figure 12b, an additional laser notch was used to force a crack initiation in the center of the notch root. The contour of the first overload (N = 15,000) shows this crack having a length of 0.7 mm and a depth of 0.2 mm. After 30,000 cycles, an additional crack has formed at the backside of the sample. The formation of both cracks is visible in the run of the potential quotients $Q_{Front}$ and $Q_{Back}$. The formation of a crack in the center of the notch root leads to a decrease of both potential quotients starting from around 11,000 cycles. The formation of the secondary crack is obviously initiated by the application of the overload at 15,000 cycles and results in a separation of the previously congruent potential curves: the potential quotient $Q_{Back}$ decreases less than the quotient $Q_{Front}$.

The investigations have shown that the multiple potential drop measurement allows an early detection of crack initiation in SEN-specimens. The calculation of potential ratios eliminates temperature effects and therefore enhances the sensitivity of crack detection. Additionally, the formation of secondary cracks can be detected via this method. In addition to the detection of crack initiation, this method also gives information about the crack front geometry of long cracks.

## 4. Conclusions

The DC potential drop method is a simple, reliable, and inexpensive method for crack length measurement in fatigue experiments and therefore accessible to many users. Despite this ease of use, some special features of this measurement method must be considered to avoid errors. Investigations on ferromagnetic materials or the occurrence of contact bridges can cause problems in automated, crack length-controlled tests and require a special processing of the measured signals.

Advances in measurement technology and the associated possibility of using amplifiers of the control electronics for the potential drop measurement enable measurements with multiple potential probes. First experiments presented here on different sample geometries have shown some possibilities arising from multiple DCPD-measurements which must be expanded and refined in further studies. Criteria must be developed for a reliable detection of crack initiation sites and the determination of the crack front geometry. The compensation of external influences by the calculation of quotients of the measured potentials opens the possibility to use this method in experiments with changing temperatures and allows the use of the multiple potential measurement in structural health monitoring.

**Funding:** This research received no external funding.

**Acknowledgments:** The experimental work in the field of multiple potential drop measurement of Levke Wiehler (SEN-specimens) and Moritz Hartweg (round specimen) is gratefully acknowledged.

**Conflicts of Interest:** The author declares no conflict of interest.

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
