# Peer review of "Crack Detection and Crack Length Measurement with the DC Potential Drop Method–Possibilities, Challenges and New Developments"

_applsci, doi:10.3390/app10238559_

Round 1

Reviewer 1 Report

Dear author

Your paper is interesting, well written and almost perfect. The only change required is figure 8a, which is empty.

Best regards,

the reviewer

Author Response

Dear reviewer,

thank you very much for your review. I have to appologize for the missing figure, it was lost by converting the manuscript to a pdf-file. I made some improvements in the manuscript and removed some grammar errors. please find enclosed the improved manuscript.

Kind regards,

Jürgen Bär

Reviewer 2 Report

The author presented the experimental setup and calibration method to measure the crack length in specimens under cyclic loading conditions using the direct current potential drop method. Besides, methods to reduce measurement errors caused by the influence of temperature and ambient atmosphere (aluminum alloys), and the Villari effect (ferromagnetic alloys) are presented. Then, the multiple potential drop measurement method is presented. The location of crack initiation on notched round bars (made of a high-alloy steel X8CrNi18-9) can be determined by analyzing the three relative potentials to evaluate the influence of the potential probe location on the measured potential. On single edge notched specimens (made of an aluminum alloy 7475 T761), three crack initiation scenarios are considered. With each initiation site, a change of the potential quotients is measured, and from which the formation of subsequent cracks is described. The multiple potential drop measurement method is proved applicable to determine the location of relatively short cracks on notched round bars and might early detect the crack initiation of edge notched specimens.

The manuscript is interesting and well-developed. The reviewer recommends for possible publication after minor revision by addressing the following comments.

  1. The author should add Figure 8a.
  2. Many grammar errors can be found in the manuscript, for example: For practical use some special … (line 10 – there should be a comma); … pressed-in pins are used the sensitivity … (line 48 – there should be a comma); This can be avoided, when wires are … (line 50 – consider removing the comma); … drop with an adaquate accuracy (line 52 – “accuracy” is the uncountable noun); … all data like force, stroke and potential (line 54 – there should be a comma after “stroke”); and many more. The errors mainly due to breaking the English punctuation rules and English articles (a, an, the). The author should revise the manuscript carefully to improve the quality of the work.

Author Response

Dear reviewer,

thank you very much for your review and your helpful comments. I have to appologize for the missing figure, it was lost by converting the manuscript to a pdf-file. I made the suggested improvements in the manuscript and removed the grammar errors. Please find enclosed the improved manuscript.

Kind regards,

Jürgen Bär
